# The world seems different in a social context: A neural network analysis of human experimental data

Maria Tsfasman[1,2☯*], Anja Philippsen[3☯], Carlo Mazzola[4,5], Serge Thill[6], Alessandra Sciutti[7], Yukie Nagai[3,8]

**1** Interactive Intelligence Group, Delft University of Technology, Delft, Netherlands, **2** Artificial Intelligence Department, Radboud University, Nijmegen, Netherlands, **3** International Research Center for Neurointelligence (WPI-IRCN), The University of Tokyo, Tokyo, Japan, **4** Robotics, Brain and Cognitive Sciences Unit, Istituto Italiano di Tecnologia, Genova, Italy, **5** DIBRIS, Università di Genova, Genova, Italy, **6** Donders Institute for Brain, Cognition, and Behaviour, Radboud University, Nijmegen, Netherlands, **7** Cognitive Architecture for Collaborative Technologies Unit, Istituto Italiano di Tecnologia, Genova, Italy, **8** Institute for AI and Beyond, The University of Tokyo, Tokyo, Japan

☯ These authors contributed equally to this work.
* MTsfasman@tudelft.nl

**Data Availability Statement:** All code and data files will be available from the github database (URL: https://github.com/mmtsfasman/TheCentralTendency_model.git).

## Abstract

Human perception and behavior are affected by the situational context, in particular during social interactions. A recent study demonstrated that humans perceive visual stimuli differently depending on whether they do the task by themselves or together with a robot. Specifically, it was found that the central tendency effect is stronger in social than in non-social task settings. The particular nature of such behavioral changes induced by social interaction, and their underlying cognitive processes in the human brain are, however, still not well understood. In this paper, we address this question by training an artificial neural network inspired by the predictive coding theory on the above behavioral data set. Using this computational model, we investigate whether the change in behavior that was caused by the situational context in the human experiment could be explained by continuous modifications of a parameter expressing how strongly sensory and prior information affect perception. We demonstrate that it is possible to replicate human behavioral data in both individual and social task settings by modifying the precision of prior and sensory signals, indicating that social and non-social task settings might in fact exist on a continuum. At the same time, an analysis of the neural activation traces of the trained networks provides evidence that information is coded in fundamentally different ways in the network in the individual and in the social conditions. Our results emphasize the importance of computational replications of behavioral data for generating hypotheses on the underlying cognitive mechanisms of shared perception and may provide inspiration for follow-up studies in the field of neuroscience.

**Funding:** This research was supported by JST CREST 'Cognitive Mirroring' (Grant Number: JPMJCR16E2), Institute for AI and Beyond at the University of Tokyo, and World Premier International Research Center Initiative (WPI), MEXT, Japan and by the Starting Grant wHiSPER (G.A. No 804388) from the European Research Council (ERC) under the European Union's Horizon 2020 research and innovation programme. The funders had no role in study design, data collection and analysis, decision to publish, or preparation of the manuscript.

**Competing interests:** The authors have declared that no competing interests exist.

# 1 Introduction

Prediction is a fundamental function of the human brain underlying various cognitive functions [1, 2], including visual perception [3]. Learning about the world by collecting experience helps us to process incoming visual stimuli in a more cost-effective manner, as we can reuse previous observations to make sense of new sensations. Predictive coding [4, 5] is a widely accepted neuro-cognitive theory that aims to explain human cognitive functions by prediction making. It claims that perception and sensorimotor responses stem from the brain's ability to constantly generate predictions about its environment and the internal states of the body. Substantial neuro-physiological evidence is consistent with the interpretation that prediction inference happens at all levels of perception [6]. It seems that most actions can be explained as aimed at minimizing prediction error: from learning basic skills [7] to interacting with peers [8].

The main assumption of the predictive coding theory is that humans use near-optimal Bayesian inference, and draw their motor-sensory decisions from combining sensory information with prior experience. They then update their prior distribution with the new information and use the updated prior distribution for generating the next prediction about the world. In Bayesian inference [9], the posterior perception depends not only on the values of the sensory and prior perceptions, but also on the precision of these signals. Specifically, signals with low variance (i.e. high precision) affect the posterior more strongly whereas signals with a higher variance (i.e. a lower precision) are less taken into account (see Bayesian inference module in Fig 1). This integration of prior and sensory information, depending on the relative precision of these two signals, improves the robustness to noise in the environment.

Central tendency, also known as context dependency or regression to the mean [10], is a well-known perceptual phenomenon revealing the use of prior experience in perception and refers to the human tendency to generalize their perceptual judgments towards the mean of the previously perceived stimuli. This phenomenon has been recently explored in the field of visual and auditory perception of time intervals [9, 11–13] and spatial distances [14, 15]. It has been demonstrated that following Bayesian criteria in the integration between information coming from prior experience and sensory stimuli in a near-optimal way well accounts for human behavior [9, 11]. For instance, to test the central tendency effect in spatial perception, participants were asked to estimate and reproduce the distances between two points. Results showed that participants tended to reproduce them closer to the average length that they perceived across trials. In other words, they underestimated longer stimuli and overestimated

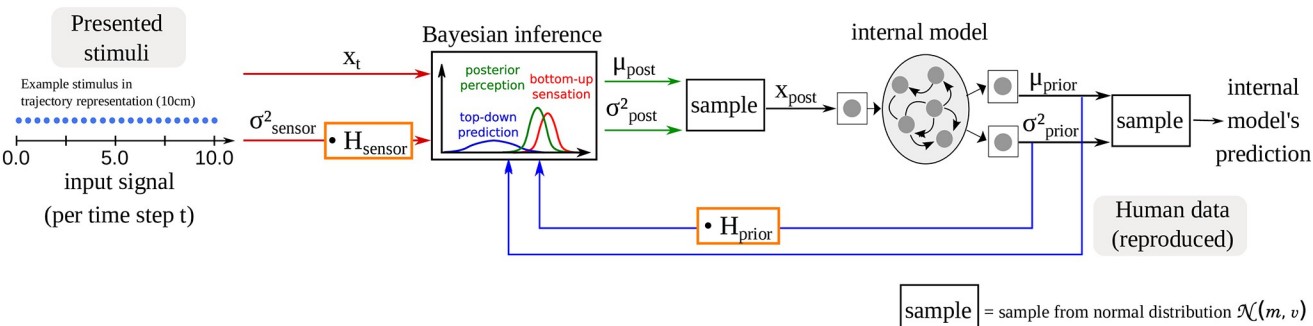

**Fig 1. An overview of the computational model used in the present paper: A recurrent neural network serves as the internal model that learns to predict future time steps of a one-dimensional trajectory whose length represents the length of the stimuli.**

shorter ones. The degree to which a person gravitated towards the mean differed between individuals.

Recently, the phenomenon of central tendency has been connected to predictive coding theories, suggesting that differences in the relative reliance on priors compared to their reliance on sensory information could account for the central tendency effect [16, 17]. The closer subjects tend to the mean, the more they relatively rely on prior experience (high central tendency); the closer they stick to the specific sensory input, the lower is their relative prior reliance (low central tendency). Previous studies have shown that central tendency is dependent on the developmental stage of a person [12, 14]. Interestingly, however, it also has been demonstrated that the extent to which humans generalize towards the mean can change rapidly depending to the context, such as whether a person is completing the task alone (or with a mechanically behaving, non-social agent) or together with another agent [15]. In particular, when interacting with a social human-like robot, participants exhibited lower central tendency and produced sensory stimuli more accurately than when they interacted with the same robot behaving mechanically, and than when they performed the task by themselves [15].

As social agents, human perceptual processes are inherently shaped by social interactions. For instance, humans engage in joint attention with co-attendants since childhood [18–21], a phenomenon which has been suggested to be at the basis for the development of perspective-taking ability, that is, the ability to intuit another person's perception, perspective, attitudes, knowledge, and so on [22]. Furthermore, sociality impacts gaze movements [23], memory processes and information encoding at different levels [23–27]. It also affects the processes of perception-action underlying joint-action [28, 29], and the adoption of different game strategies [30]. Finally, it influences perception of space [15, 31, 32].

The effect of the social context on perception found in [15] indicates that social interactive context might modify perception and, specifically, the relative reliance on prior and sensory information while processing perceptual information. It is known that sociality widely affects behavior and cognition [33–36]. While the exact mechanisms are still unknown, neurobiological mechanisms such as neuromodulators could link the social context to perception and behavior. Specifically, there is evidence that neuromodulators which play an important role in social behavior [36] can alter perception, for instance, in the example of psychosis [5, 37]. Social interactions could increase the level of dopamine and, as a result, modulate the precision between sensory and prior information [5, 38]. An alternative mechanism is attention. Social interaction could highlight signals from the external world and, thus, highlight the sensory information in contrast to prior information.

A major problem in investigating social effects is to isolate the effect of sociality in experimental studies. Manipulating experimental conditions does not only affect the sociality but also has many side-effects such as differences in the attention and cognitive load of the participants. The question about the underlying neural mechanisms cannot be easily answered using a behavioral experiment since it would require an analysis of the neural activation of the human brain—a challenging task given its complexity. However, one way to investigate the potential underlying mechanisms of the observed behaviors is by using a computational model that replicates the human behavioral data using a simplified neural system—an approach that is commonly used to investigate broad behavioral phenomenons which lack clear hypotheses applicable at a neural level [39]. Such neural network approaches which replicate the human behavioral data using a simplified neural system may provide a tool for exploring the role of various neural mechanisms on human perception and generating new hypotheses to be tested in neurobiological as well as psychological studies.

From this perspective, here, we train an artificial neural network on the human experimental data from [15] to better understand the neural mechanisms underlying the context-

dependent variation of reliance on the prior that were found in the human behavior. Specifically, we are interested in the mechanisms that play a role in how humans differentiate between individual and social task conditions. The neural network we use for this purpose was originally introduced in [40] and integrates a recurrent neural network model that learns to make predictions about the world, functioning as an internal model, and a Bayesian inference module that combines sensory input and the predictions of the internal model based on the precision of these two signals. We conduct two experiments using this model. In the first experiment, we manipulate the hyperparameters of the model to modify the network's reliance on sensory and prior information. This allows us to investigate how such alterations affect the behavioral output of the network. Second, we analyze the neural dynamics that emerge in the neural network during training to evaluate which mechanisms the network might be using to differentiate the three conditions using its neural encoding.

Using this design, we aim at answering the question which neural mechanisms might explain the change of the reliance on the prior and sensory signals found in the social condition. Our hypothesis is inspired by the Bayesian perspective on predictive coding that behavioral differences may be caused by an altered precision of the sensory and the prior signals. For example, a more precisely perceived stimulus would cause a sharper perception and, consequently, a higher reliance on the sensory input when performing perceptual inference. We demonstrate that changes in the precision of prior and sensory information can replicate the central tendency effect as hypothesized in previous studies [16, 17], and discuss implications that this could have on social perception based on the results of [15].

## 2 Background

In this section, we first explain the human behavioral experiment conducted by [15] in detail. We then introduce the computational model used here, describe the training procedure and confirm that it is able to replicate the human data.

### 2.1 Summary of the human behavioral experiment

Mazzola and colleagues [15] investigated whether the level of social involvement into the task can affect the perceptual phenomenon of central tendency that had already been explored in previous experiments and described in Bayesian terms [9, 11, 12, 14]. The central tendency effect refers to a phenomenon where, given a series of stimuli of the same type, the perception of one stimulus is influenced by the stimuli perceived before. Specifically, when the observer has to reproduce the magnitude of a stimulus (here, the spatial distance between two points), the reproduction gravitates to the average of all the stimuli perceived before. Thus, in line with predictive coding theories, the reproduced length is affected both by sensory information and the participant's prior, with the balance between these two signals determined by their individual precision.

In [15], participants were exposed to visual stimuli of different lengths and asked to reproduce them. To test how social context affects the visual perception of space, the experiment was conducted in three different conditions that only varied in the way stimuli were presented to the participants:

1. **Individual condition**: Participants performed the reproduction task by themselves. In each trial, two points indicating the endpoint of the stimulus were subsequently shown on a tablet touch screen. After the last point disappeared, the participant had to reproduce the length of the stimulus by touching the screen at a distance from the last point equal to the distance between the two presented points.

2. **Mechanical robot condition**: The same task as in the previous condition was used but now the endpoints of the lengths were indicated by a humanoid robot iCub [41] that touched the tablet in front of the participant with its index finger. Throughout the whole task, the robot appeared as a mechanical agent: it looked away from the participant and did not produce any additional verbal or non-verbal cues.

3. **Social robot condition**: The same setup as in the mechanical robot condition was used, except that the behavior of the robot was modified to appear more social and human-like. This included saying hello to participants and explaining them the task, making eye contact, smiling and uttering encouraging phrases.

The individual condition was used as a baseline condition to measure the central tendency of the participants. The mechanical and social robot conditions were designed to control them for potentially confounding factors such as the cognitive load of the participants or differences in the spatial distances. The task design of these two conditions was identical and only differed in factors that were directly related to the experimental manipulation affecting the "socialness" of the situation.

During the task, a series of 11 different lengths from 6 to 14 cm was shown to participants, each series repeated 6 times throughout the task in a randomized order. While in the individual condition the lengths were shown by two dots appearing on the screen, in the other two conditions, these two points were shown directly by the robot touching the screen with its right index finger to foster the interaction between participants and iCub. The contact between the robot's finger with the screen caused though some imprecision in the stimuli demonstration. Therefore, in some trials, the touch screen did not successfully feel the touch of iCub so that in the final data set there is a mean of 62.66 trials (SD = 3.83) for the robot condition. Also, the distances recorded as stimuli in the robot conditions slightly differed from the ones of the individual condition since they were not computed as the ideal lengths sent to the robot, but as the real ones recorded by the touch screen after the touch of the robot (mean variation: M = 0.62 cm, SD = 0.13). In this way, it has been possible to record what had been factually seen by participants and give a more precise measure of the regression index of participants. Considering the phenomenon of central tendency, the regression index is a measure of the degree to which participants tend towards their prior [11, 12, 14, 15], where the prior is calculated as the average of all the stimuli perceived during one condition, while the regression index is computed as the difference in slope between the best linear fit of the reproduced values plotted against the corresponding stimuli and the identity line, which would correspond to the ideal perfect perception of the exact lengths of the stimuli. Thus, a regression index close to 1 reveals a strong influence of priors, while a regression index close to 0 reflects a weak influence of the prior and a strong tendency to perfectly reproduce the presented stimulus length. All the participants gave their written informed consent before participating. The regional ethical committee approved the study (Comitato Etico Regione Liguria).

Mazzola and colleagues [15] found that the reliance on priors was stronger in the individual task compared with the two robot conditions. As the difference of the individual to the robot conditions might have been caused by confounding factors, importantly, the results also revealed a variation in human perception between the two conditions with the robot: participants were less influenced by their priors when performing the task with the social robot and thus reproduced the stimuli more accurately. This change was only induced by the modification of the socialness of the situation. That the experimental manipulations actually affected the social perception of the participants could be confirmed via the scores of an anthropomorphism questionnaire filled out by the participants which indicated that the more participants perceived the robot as human-like, the higher was the difference of the regression index

between the two conditions, resulting in a greater accuracy for the condition with the social human-like robot. For a recent paper providing more in-depth analyses of the experimental results see [42].

## 2.2 The computational model

The computational model used in this study is made of two components: a stochastic continuous-time recurrent neural network (S-CTRNN) [43] that serves as the internal model which learns to make predictions about the world, and a Bayesian inference (BI) module that integrates sensory input with the priors generated by the internal model. This network model was first presented by [40] and was used to predict how people and chimpanzees would perform a drawing completion task in [44]. We chose this particular model since it both follows the principles of predictive coding and allows us to modify the precision of the model's prior as well as the precision of sensory perception.

The S-CTRNN network is able to recurrently predict the mean and the variance of the next time step of a time-dependent signal, where the mean is the estimated next values and the variance expresses the uncertainty of this estimation. As a higher variance means that the precision of the signal is lower, and vice versa, the estimated variance may also be described as inverse precision. Formally, given input $\mathbf{x}^t$, the S-CTRNN predicts the mean $\mu_{prior}$ and the variance $\sigma^2_{prior}$ of the sensory perception of the next time step $\mathbf{x}^{t+1}$ (Following standard conventions, we denote scalars as $x$ and vectors as $\mathbf{x}$. However, note that for this experiment, the input dimension $D = 1$.).

The context layer consists of 25 neurons which we found to be sufficient for well learning the 1-dimensional task. All network connections are linear mappings with weights and no bias terms. The input is mapped to the context layer via weights $\in \mathbb{R}^{1 \times 25}$, recurrent weights are defined $\in \mathbb{R}^{25 \times 25}$, and the context layer is mapped to the mean and to the variance output unit, respectively, via a weight matrix $\in \mathbb{R}^{25 \times 1}$.

To train the network to reproduce human behavior, the backpropagation through time algorithm is used as described by [43]. Specifically, during training, which proceeds in epochs, the likelihood that the output mean and output variance of the network resembles the human data is maximized by updating the network weights. In other words, the prediction error, scaled by the estimated variance, is minimized.

The likelihood $L$ that is maximized consists of two terms $L = \ln L_{out} + \ln L_{init}$. $L_{out}$ is the likelihood that the network's estimated mean $\mu_{prior}$ and variance $\sigma^2_{prior}$ account for the observed input x:

$$\ln(L_{out}) = \sum_{t=1}^{T} \sum_{i=1}^{D} \left( -\ln\left(2\pi\sigma^2_{prior}t, i\right) - \frac{(x^{t+1,i} - \mu^{t,i}_{prior})^2}{2\sigma^2_{prior}t, i} \right), \tag{1}$$

where $T$ is the total number of time steps (here, $T = 22$), and $D$ is the dimensionality of the input vector (here, $D = 1$).

The term $L_{init}$ is used as introduced in [43] and optimizes the distance between the initial activations of the recurrent layer, the so-called initial states:

$$\ln(L_{\text{init}}) = \sum_{s=0}^{S-1} \sum_{n=0}^{N-1} \left( -\ln\left(2\pi\sigma^2_{init}\right) - \frac{(u^{0,n}_{(s)} - \hat{\mathbf{u}}^n)^2}{2v_{\text{dist}}} \right), \tag{2}$$

where $N$ is the number of neurons in the context layer (here, $N = 25$) and $S$ is the number of initial states that the network should differentiate (here, one initial state per condition and per

participant is used, resulting in $S = 3 \cdot 25 = 75$ initial states). $u^{0,n}_{(s)}$ refers to the initial state (e.g., neuron activation vector at time step $t = 0$ for the $s$-th initial state of the $n$-th neuron. $\hat{\mathbf{u}}^n$ is the (learnable) mean of all initial states and $v_{\text{dist}}$ (set here to $\sigma^2_{init} = 1e7$) is the predefined variance of the initial states.

Initial states are required because the S-CTRNN is a deterministic system, therefore, given one set of activations of the recurrent layer neurons and a specific input signal, the network would, once it is trained, always produce the same output. However, we want to train the model to replicate the behavior of various study participants in different experimental condition in a single model, such that we can directly compare between the way that different conditions and different participants are represented in the neural system. By representing different participants and different conditions with different initial states, the separation of different types of behaviors within the network dynamics can be achieved automatically during the training process. In this way, different participants and different conditions can be represented in the network with different neural dynamics, while reusing the same neurons and weight matrices. Note that summarizing all participants and conditions in a single neural network is done here for convenience purposes to easily compare and switch between the conditions to fairly compare between the neural representations of different conditions and participants independently of confounding network parameters as they may result from training. Importantly, we do not intend to use the neural network here as a biologically plausible simulation of the participant's brain but instead as a computationally tool. Specifically, the network is provided with the information of which training trajectory belongs to which initial state during training. Using the two likelihood terms, the network gradually differentiates the initial states during training. $L_{init}$ defines a target variance $\sigma^2_{init}$ that determines the desired variance between different initial states (see [43] for details). Generally, a higher variance between initial states leads to a stronger separation of the neural dynamics of different participants and conditions.

At each time-step, the output mean and variance predicted by the internal model is fed into the Bayesian inference module where it is combined with the raw sensory input and the corresponding precision (Fig 1) depending on the ratio of sensory and prior precision. Specifically, the mean and the variance of the posterior distribution is calculated as:

$$\sigma^2_{post} = \frac{(H_{\text{sensor}} \cdot \sigma^2_{sensor}) \cdot (H_{\text{prior}} \cdot \sigma^2_{prior})}{(H_{\text{sensor}} \cdot \sigma^2_{sensor}) + (H_{\text{prior}} \cdot \sigma^2_{prior})}, \tag{3}$$

$$\mu_{post} = \sigma^2_{post} \cdot \left( \frac{\mu_{prior}}{(H_{\text{prior}} \cdot \sigma^2_{prior})} + \frac{x}{(H_{\text{sensor}} \cdot \sigma^2_{sensor})} \right). \tag{4}$$

The distinguishing feature of this computational model is that it allows us to manipulate the reliance on the prior and the sensory signal via parameters $H_{\text{prior}}$ and $H_{\text{sensor}}$ to simulate a stronger or weaker reliance on either the prior or the sensory input. These two parameters function as a factor that is multiplied with the variance of the prediction $\sigma^2_{prior}$ or with the variance that is associated with the sensory signal $\sigma^2_{sensor}$.

During training of the network, $H_{\text{prior}} = H_{\text{sensor}} = 1$ is used such that the network learns to correctly replicate human data. Different conditions of $H_{\text{prior}}$ and $H_{\text{sensor}}$ during training are not considered here because we are interested in the relative changes of the network's behavior when switching between the different conditions, and not on the effects of the parameters on the developmental processes. These parameters can later on be changed to higher or lower values to modify the reliance of the model on prior or sensory signals. Specifically, choosing $H_{\text{prior}}$

$> 1$ increases the expected variance of the prior, leading the network to rely less on the prior. In contrast, choosing $H_{\text{prior}} < 1$ decreases the variance and causes the network to rely more on its learned prior while performing the task. $H_{\text{prior}}$ and $H_{\text{sensor}}$ can be set independently from each other to increase or decrease the precision of either prior or sensory information. Both affect the ratio between the precision of sensory and prior information and, thus, have comparable effects on the model (an increase of prior precision has similar effects as the decrease of sensory information). Still, the effect of both parameters is investigated here as also the absolute values affect perception, namely they determine the variance of the posterior. For example, if the precision of both signals is low, the posterior mean would be the same, but the variance is much higher than when both signals are rather precise, even when the ratio between sensory and prior precision is the same.

## 3 Training the model to replicate human data

The main goal of this study was to verify whether the differences between individual and social perception between experimental conditions can be replicated by continuous modification of one parameter (e.g, prior reliance), and whether there might be multiple mechanisms causing behavioral differences. As a first step to investigate these issues in Sections 4 and 5, first the model has to be trained with the human experimental data from [15]. In this section, we describe how the network was trained and verify that the performance of the network replicates human performance with sufficient accuracy.

### 3.1 S-CTRNN training

In contrast to [40, 44] where the network was trained to directly reproduce the presented input trajectories (i.e. input equals output of the network), we train the network by providing the stimuli presented to human participants as input while the output corresponds to the participants' reproduction of these stimuli. As such the training mimics human learning of the task as closely as possible.

The network was trained with all the data from the human experiment which involves the data of 25 participants who performed the task in three different conditions.

**3.1.1 Training data.** The training data were taken from the behavioral experiment of Mazzola and colleagues [15] as described in Section 2.1. Since the S-CTRNN model is designed to learn the next time-step of trajectories, the lengths from the human data had to be modified into one-dimensional trajectories consisting of multiple time-steps. Each trajectory started at location 0 and ended at the particular length of the stimulus. 20 equally spaced points were inserted between the start and the end points of each line, resulting in trajectories consisting of 22 time steps. An example is shown for a stimulus of 10cm on the left side of Fig 1. Before using the data for network training, the trajectories were normalized such that all trajectory points fall within the range $[−1, 1]$. Hence, after normalization, all trajectories start at $−1$. The representation of stimuli as a trajectory alters the setting from the human experiment where participants just pointed at the final position, but including intermediate points may also provide new opportunities. As we will see later in Section 5, this design allows us to look into the length reproduction task as a dynamical process.

Both the presented stimuli and the lengths reproduced by participants were converted into multi-step trajectories in the same way. While the presented trajectories served directly as network input, the reproduced lengths were used for the prediction error computation during network training.

**3.1.2 Training parameters.** As motivated above, the model had different initial states for each participant and condition, resulting in $75 = 25 \cdot 3$ initial states. These initial states were

automatically determined during training, using a high maximum initial state variance ($\sigma^2_{init} = 1e7$) to ensure that the neural dynamics of different conditions and participants are sufficiently separated from each other.

The parameters $H_{\text{prior}}$ and $H_{\text{sensor}}$ were set to 1 during network training while the number of neurons in the recurrent network layer was set to 25. The network was trained for 15000 epochs.

Ten networks were trained independently from each other, using different randomly chosen sets of initial weights. By investigating the performance of a set of networks, we can ensure that the results that we find are reliable and not caused by random effects.

**3.1.3 Network behavior generation.** Similar to the way that the human experiment was conducted, we tested the performance of the network by providing it with trajectories of different lengths. This test set corresponded to the data that was presented to the human participants. To generate the behavior for a specific participant and experimental condition, the corresponding initial state of the network was used to initialize the activations of the recurrent network layer. Then, the network's output, given the input, was computed to generate the model's behavior. From the trajectories that the network produced in response to the presented stimuli, the reproduced lengths were computed as the absolute difference between the start and end points of the reproduced trajectory. A linear model was fit to the reproduced lengths in order to compute the regression index. Additionally, the neural activation history of the recurrent layer was recorded and used for neural representations analysis in Section 5. The resulting neural activation data consisted of the activation for each neuron of the model for each trajectory time-step for each trajectory for each participant and condition.

## 3.2 Network performance

A comparison between the human experimental data and the performance of a trained network for six randomly chosen participants in the three different conditions is presented in Fig 2. The x-axis shows the presented lengths, the y-axis the length reproduced by the human participants (left) or by the model when using the corresponding initial state (right). Lines in the right plot show the result of the linear regression that was performed in order to calculate the regression index.

It can be observed that the model is able to accurately replicate the mean of the human data. Note that for generating the results in this figure only the mean without the uncertainty was generated by the model to get a better impression of the model's behavior. Therefore, the variability of the human data is not replicated on the right side of Fig 2.

A direct comparison of the regression indices of the model behavior with the corresponding regression indices of the human behavior is shown in Fig 3 including the data of all ten networks. It can be observed that the model's behavior slightly diverges from human behavior, however, the large majority of stimulus replications accurately correspond to the regression index of the corresponding human participant. It can also be seen that in the individual condition, a stronger regression towards the prior is taking place than in the other conditions in the human data as well as in the model data.

Black dots in Fig 4 show the subject-wise difference between individual–mechanical, individual–social and mechanical–social conditions, an important measure to visualize differences between conditions also used by Mazzola and colleagues [15]. This distance is the highest for individual–social, indicating that the regression index is significantly higher in the individual condition compared to the social condition. The mechanical–social difference is smaller, but significantly higher than zero, indicating that the regression indices of the mechanical and the social condition lie closer together.

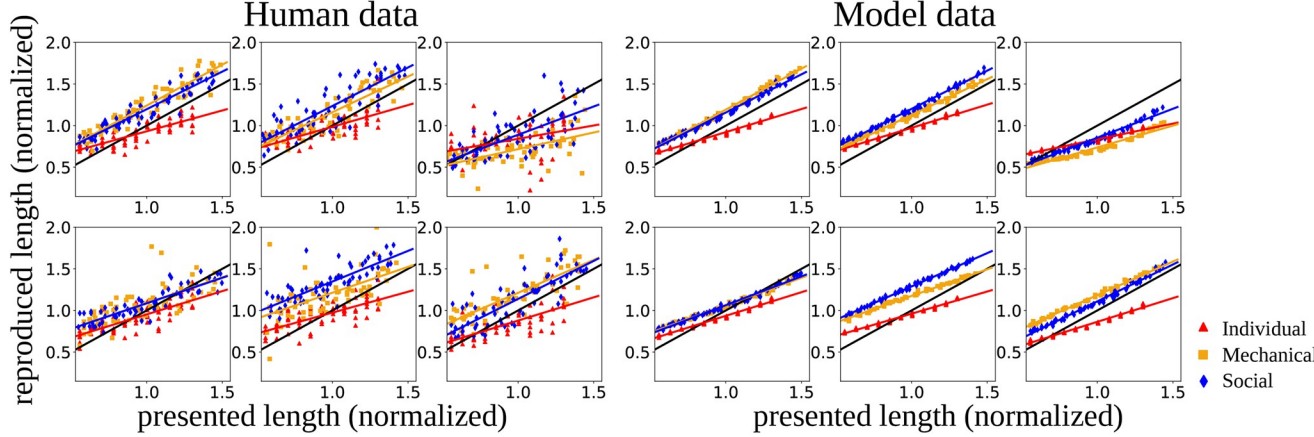

**Fig 2. The reproduced lengths plotted against the presented lengths, where lengths were calculated in the normalized space of trajectories.** Original human data (left) is compared with the corresponding mean predictions produced by one example network (right) for six randomly chosen participants. Lines in both plots correspond to the regression lines extracted from the human data or the model data, respectively. The black line shows the identity line.

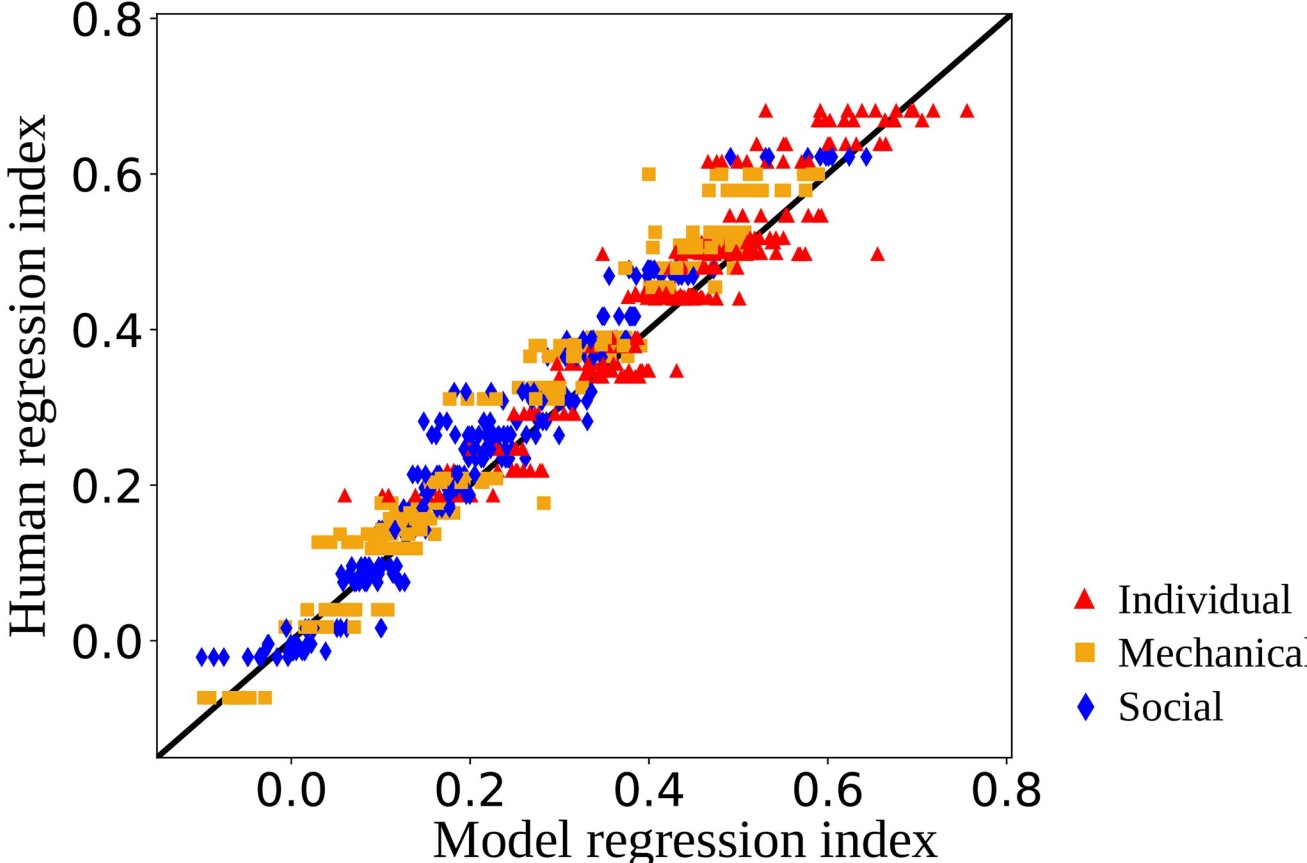

**Fig 3. The regression indices of the human plotted against the regression indices of all trained networks for reproducing all training data.**

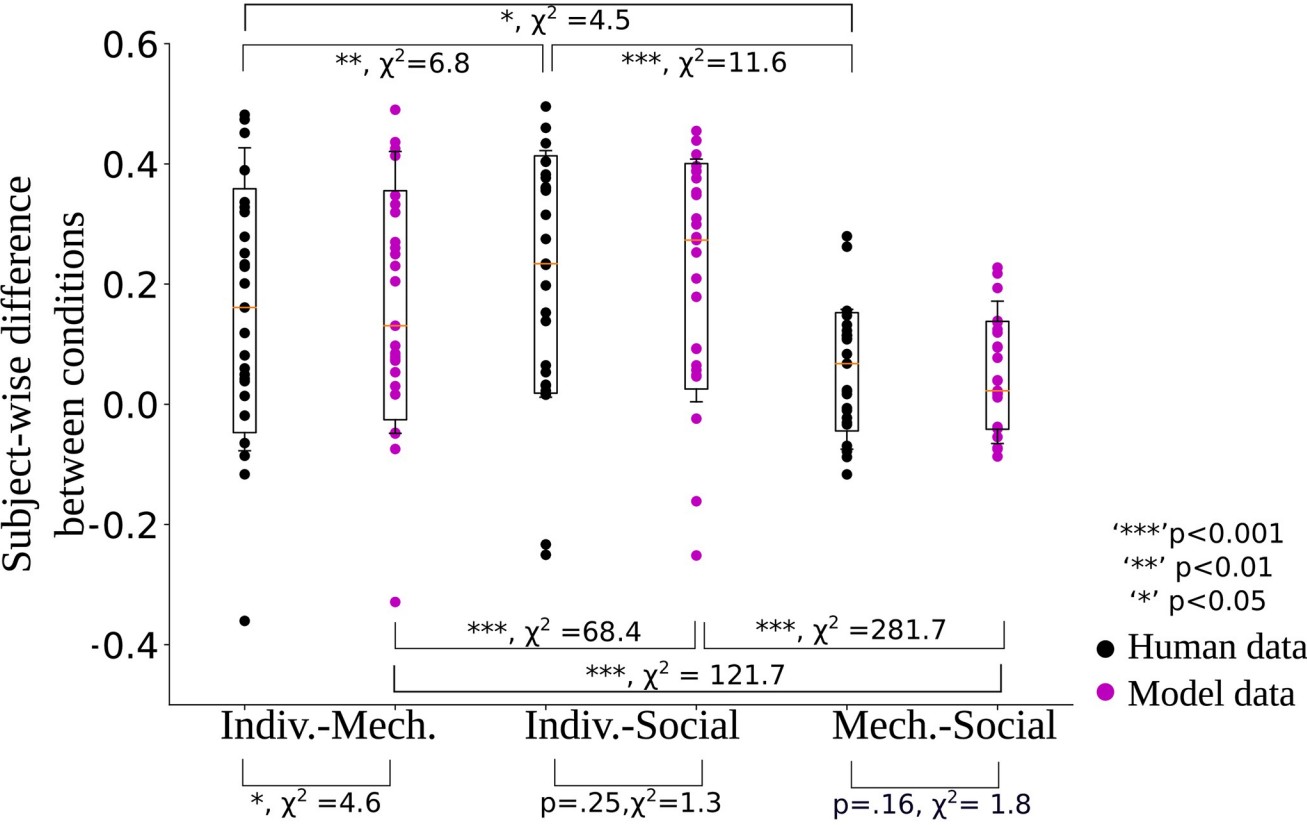

**Fig 4. Subject-wise differences between different conditions, compared for human data (black) and model data (magenta) for one trained example network.** Boxes indicate the mean, and 80% confidence intervals of the data, fliers indicate standard deviation. Model data reproduce the main trends of the data, but with slightly lower variability. The p-values were computed using the results of all ten networks, i.e. on 25 samples from the human participants, and 250 (= 10 · 25) samples from the models.

Purple dots in Fig 4 show the same analysis conducted for the model results. It can be observed that the trends in the model behavior well replicate the human behavior, but the variability is slightly reduced in the model data compared to the human data. Specifically, the standard deviation of the model data is on average 7% smaller than in the human data. Furthermore, there is a small significant difference between the model and the human data in the individual–mechanical condition difference.

The p-values, computed on all ten networks, are shown in Fig 4 in detail and were determined using linear mixed effect models, describing the subject-wise difference by either the conditions (e.g. individual–mechanical vs. individual social) or by the agent (i.e. human vs. model) with the subject ID as a random effect.

Overall, this analysis demonstrated that the model is able to replicate the important trends that are present in the human data. Based on the trained models, we conducted two sets of analyses we call here experiment 1 (section 4) and experiment 2 (section 5). Experiment 1 aims to answer the question whether it is possible to replicate the human results in different conditions with a continuous change of one parameter in the model. In short, experiment 1 looks at how the model performs in the length reproduction task depending on its prior reliance. Experiment 2 investigates how the differences between conditions are represented in the neural activations of the network. It allows us to look deeper into the mechanisms behind the differences in model performance and verify whether there are other processes at stake.

## 4 Experiment 1: Changes in the reliance on prior and sensory information

In the human experiment [15], it was found that participants tended more strongly towards the prior in the individual condition, and more accurately replicated the stimuli in the social robot condition, while the mechanical robot condition lied in between. This finding suggests that there might exist a continuum between the three conditions from the individual condition to the social condition via the mechanical condition.

The parameters $H_{prior}$ and $H_{sensor}$ of the computational model we are using here (see Section 2.2) can be used to implement such a continuous change as they modify the ratio to which sensory information and predictions are integrated while replicating the perceived lengths.

In this section, we test the hypothesis that a continuous change of $H_{prior}$ or $H_{sensor}$ respectively can replicate changes in the human behavior between the individual, mechanical robot, and social robot condition. We first modify only $H_{prior}$ in Section 4.1; then, we test whether modifying $H_{sensor}$ has analogous effects (Section 4.2).

### 4.1 Experiment 1A: Modifying the reliance on prior predictions

In the human experiment, the weakest reliance on the prior was found in the social robot condition. Therefore, our expectation is that when gradually increasing the model's reliance on the prior, a network behavior that was formerly replicating the social robot condition would produce behavioral results which would be closer first to the mechanical robot (with moderate increase of prior reliance) and then to individual conditions (strong increase of prior reliance). If this hypothesis is correct, it should be possible to find values of $H_{prior}$ such that the network behavior replicates the human behavior in the individual and mechanical robot condition, while only using the initial states of the social robot condition.

To test this idea, in this experiment, we use only a subset of the trained network dynamics, namely, the 25 initial states that are associated with the social robot condition. Then, we test whether it is possible to replicate the results of the other two conditions by adjusting $H_{prior}$.

The network's behavior was tested by using a wide range of values between 0.5 and 0.05 for the $H_{prior}$ parameter. For each of the different values of $H_{prior}$ the network behavior was recorded. Similarly to Fig 4, subject-wise differences between conditions were computed as a measure of how well the replicated lengths fit human data. Specifically, the difference was computed between the replicated length of the initial state of the social robot condition with $H_{prior} = 1$, and the replicated length of the initial state of the social robot condition with $H_{prior} = x$ where $x \in$ 0.5, 0.45, 0.4, 0.35, 0.3, 0.25, 0.2, 0.15, 0.1, 0.09, 0.08, 0.07, 0.06, 0.05. These values were selected in an iterative way based on how variable the behavior changed in a certain parameter region.

Fig 5a shows the median difference of all ten networks (different colors refer to different networks). The horizontal dashed lines in Fig 5 indicate the subject-wise difference between the social robot condition and the mechanical robot condition in the human data and the difference between the social robot condition and the individual condition in the human data. It can be observed that a stronger prior (i.e. a smaller value of $H_{prior}$) gradually increases this ratio, that is, with the increased prior reliance the produced lengths tend more strongly towards the mean of the data. The nonlinear decay is a consequence of the mathematical formulation of Bayesian inference. See appendix for further information. A value of $H_{prior} = 0.4$ closely matches the social–mechanical difference of the human data, and $H_{prior} = 0.1$ closely replicates the social–individual difference of the human data.

Fig 5b shows the subject-wise differences between conditions for a few selected values of $H_{prior}$ for the data from a single network. This plot allows us to inspect not only the median

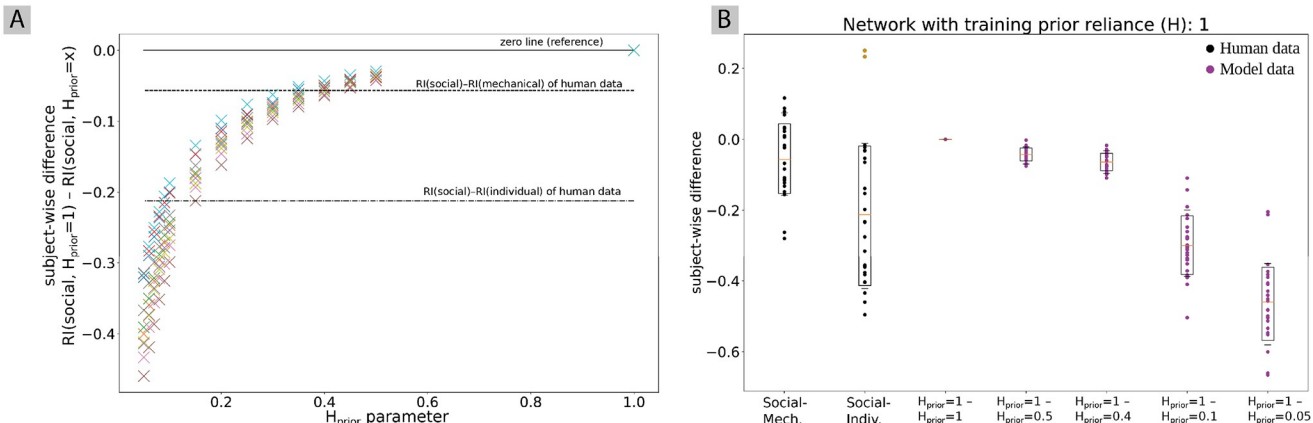

**Fig 5. Difference between the regression index of networks produced using the 25 initial states of the social condition with regular prior reliance ($H_{prior} = 1$) and the regression index produced with the same initial states using increased ($H_{prior} < 1$) prior reliance.** (a) For all ten networks the median of the subject-wise difference is displayed. Horizontal lines mark the zero line, the average subject-wise difference in the regression index between the social and the mechanical condition in human data, and the average subject-wise difference in regression index between the social and the individual condition. (b) Detailed results including all subject data for a single network. The subject-wise differences between the behavior using social initial states of $H = 1$ vs. $H = x$ for different $x$ values is displayed.

but also the variability between different participants. It can be observed that although the median for $H_{prior} = 0.4$ and $H_{prior} = 0.1$ match the median of the human data, the standard deviation is much larger in the human data. However, the further away the value of $H_{prior}$ is from the standard value of $H_{prior} = 1$, the larger the standard deviation becomes. We tested statistically whether there is a difference between the subject-wise difference reproduced by the model in the different conditions and the corresponding human data. For this purpose, we used linear mixed effect models describing the subject-wise difference as a function of the identity of the agent (i.e. whether it is human data or model data) using the subject ID and the network ID as random effects. The subject-wise difference between $H_{prior} = 1$ and $H_{prior} = 0.4$ and between $H_{prior} = 1$ and $H_{prior} = 0.1$ showed no significant difference when compared to the social–mechanical difference or the social–individual difference in human data, respectively.

The results demonstrate that it is possible to replicate the tablet and the mechanical condition using the initial states of the social condition, i.e. we can switch from weak towards strong prior reliance. Theoretically, we could also go into the opposite direction, trying to modify the network behavior by moving from a strong towards a weak prior, i.e., replicate the mechanical and the social condition, starting from the tablet condition. However, executing the experiment showed that the subject-wise differences of the tablet condition did not change regardless of the $H_{prior}$. Specifically, even when changing $H_{prior}$ to a value close to 0, the subject-wise difference remains the same (results can be found in the github repository, https://github.com/mmtsfasman/TheCentralTendency_model). The reason for this finding is that the networks were trained to replicate human data and not to replicate the actual presented stimuli. Human subjects do not have perfect precision, thus, the human data that the network was trained with also does not reflect the actual presented stimuli. Therefore, the network is not able to achieve higher accuracy than the human subjects even if the attention is shifted to the sensory signal. Demonstrating the shift from a stronger towards a weaker prior, thus, is not possible with the current experimental design. In contrast, it is always possible to shift towards a more strong prior as this does not require any knowledge about the presented stimuli but is implicitly

known in the model. Therefore, we focus in this section on demonstrating the shift from a weak to a strong prior.

## 4.2 Experiment 1B: Modifying the reliance on sensory information

Section 4.1 demonstrated that changing $H_{prior}$ can replicate the behavioral differences between the conditions. This parameter can be intuitively interpreted as the inverse precision of the network's prior. However, modifying the inverse precision of the sensory input $H_{sensor}$ could yield similar results. To test whether a change in $H_{prior}$ or $H_{sensor}$ better explain the human data, we repeated experiment 1A, modifying $H_{sensor}$ instead of $H_{prior}$. As explained above, the result of the Bayesian inference is mainly affected by the ratio of $H_{sensor}$ and $H_{prior}$, but the absolute values of the two parameters change the variance of the posterior.

To evaluate whether changes of $H_{sensor}$ equally allow us to change the behavioral output of the network according to the human conditions, we selected values of $H_{sensor}$ such that the ratio between sensory and prior precision is the same as in experiment 1A. For example, setting $H_{prior} = 0.5$ leads to a ratio between $H_{prior}$ and $H_{sensor}$ of $0.5 : 1 = 0.5$. The same ratio of 0.5 can be achieved by keeping $H_{prior} = 1$ but increasing $H_{sensor}$ to a value of 2. Thus, the corresponding value of $H_{sensor}$ that produces the same ratio as the $H_{prior}$ value that was used in experiment 1A can be computed as $H_{sensor} = H_{prior}^{-1}$.

The results are displayed in Fig 6a. Like in experiment 1A, the figure shows the median difference of all ten networks (different colors refer to different networks) between the produced lengths observed with $H_{sensor} = 1$ and with $H_{sensor}$ set to the values displayed on the x-axis of Fig 6. Again, the horizontal dashed lines indicate the difference between the social robot condition and the mechanical robot condition in the human data and the difference between the social robot condition and the individual condition in the human data. While in Fig 5a the value of H was gradually decreased to increase the reliance on the prior signal, in Fig 6a the value of $H_{sensor}$ is gradually increased to decrease the reliance on the sensory signal.

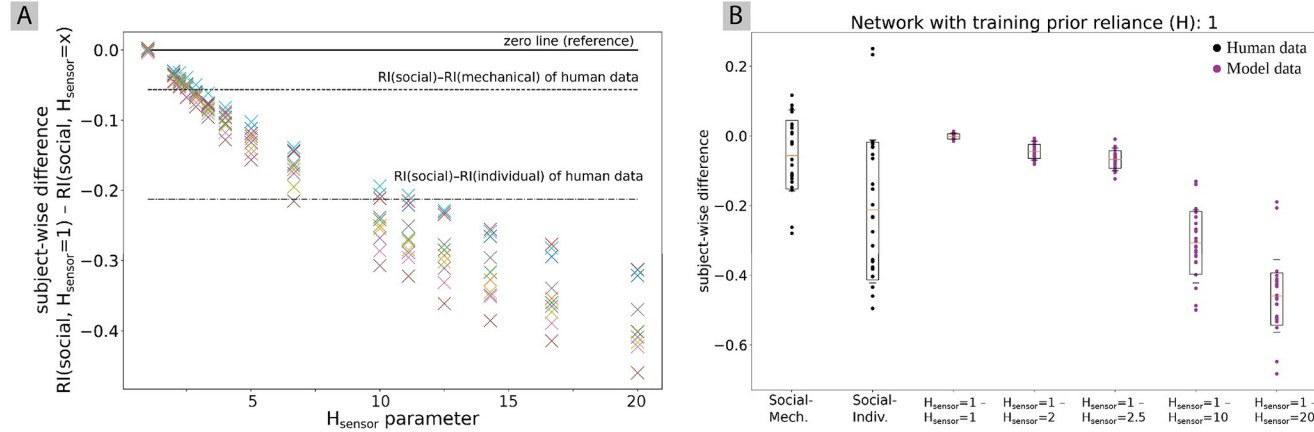

**Fig 6. Difference between the regression index of networks produced using the 25 initial states of the social condition with regular reliance on sensory information ($H_{sensor} = 1$) and the regression index produced with the same initial states using decreased ($H_{prior} > 1$) sensory reliance.** (a) For all ten networks the median of the subject-wise difference is displayed. Horizontal lines from top to bottom mark as indicated the zero line, the average subject-wise difference in the regression index between the social and the mechanical condition in human data, and the average subject-wise difference in regression index between the social and the individual condition. (b) Detailed results including all subject data for a single network. The subject-wise differences between the behavior using social initial states of $H = 1$ vs. $H = x$ for different $x$ values is displayed for $H_{sensor}$.

The results show a similar change of the difference with gradual modification of the parameter. The human data differences are replicated with $H_{\text{sensor}} = 2.5$ for social––mechanical and with $H_{\text{sensor}} = 10$ for social––individual. With these values, the exact same precision ratio between prior and sensory precision is achieved as with the corresponding values found in experiment 1A. The corresponding plot of a single network Fig 6b shows identical results to Fig 5b, indicating that in the present experiment a modification of $H_{\text{sensor}}$ or $H_{\text{prior}}$ lead to equivalent behavior changes.

We tested whether the difference between human and model data is significant for the individual parameter conditions analogously to the procedure described in Section 4.1. Also here, no significant differences were found for the above parameter values ($p > .05$), indicating that the model data well describe human data.

## 4.3 Discussion

The results of experiment 1 indicated that reliance on the prior could account for the differences we see in the behavioral differences between the three conditions. Specifically, we tested whether it is possible to gradually modify the network's behavioral output from weak prior reliance as it was found in the social robot condition of the human data towards a strong prior reliance as it was found in the individual condition of the human data. We found that a gradual shift of $H_{\text{prior}}$ as well as of $H_{\text{sensor}}$ could switch the network's behavior from the social condition to the other two conditions, indicating that all observed behaviors could be explainable based on the same underlying mechanism.

Notably, the same behavior could also be achieved by changing the reliance on sensory information instead of prior information. Further, while experiment 1A and 1B could in principle yield differences in the variances of the behavioral output, no significant difference could be observed between the two mechanisms. Thus only the ratio, not the absolute values of $H_{\text{prior}}$ and $H_{\text{sensor}}$, influenced the behavioral outcomes.

One reason why we did not find any differences depending on the absolute amplitudes of $H_{\text{prior}}$ and $H_{\text{sensor}}$ might be the fact that the task was too simple and thus easily learned by the network. A more complex encoding of the experimental data, which also takes into account the variability of the generated output could help to make differences between experiment 1A and 1B visible. Here, the variance is estimated but not explicitly modeled in the data as a sample is drawn from the estimated posterior distribution. Modifying the input encoding to explicitly model the variance of the signal, using for example population coding [45], could help to investigate whether differences between changes in prior and sensory reliance might exist. For the purpose of our investigation, however, the current implementation is sufficient as we were rather interested in the possibility to model the differences using a single parameter than in the differences between modifying prior or sensory precision.

In conclusion, experiment 1 demonstrated that a gradual change of the reliance on prior or sensory information can replicate the changes that we observed in the human data. Therefore, it seems possible that human cognition makes use of the same underlying cognitive mechanism regardless of the situational context, but modifies this mechanism along a continuum to fit situational constraints. Specifically, the precision associated with the sensory and prior signal might be modified depending on the amount of social information that is present in the experienced situation.

These experiments demonstrated that changes of the precision of sensory and prior signals might be directly connected to the observed behavioral changes. However, this is only one possible explanation. In the following subsection, we explore the alternative hypothesis, namely,

that there are fundamentally different cognitive mechanisms underlying the behavioral change observable between the three experimental conditions.

## 5 Experiment 2: Analysis of internal network dynamics

While the results obtained in experiment 1 render it plausible that the same cognitive mechanisms might underlie the behaviors observed in all conditions of the human experiment, the differences among conditions in the human experiment might be caused by fundamentally different underlying cognitive mechanisms. For example, the difference between the individual condition and the two robot conditions seems to be of different nature than the change between the mechanical and social robot condition. It is not only a change in the social, but also in the perceptual domain: whether a point simply appears on a screen or is indicated by a moving robot affects the whole experience of the participant. The difference between the mechanical and the social robot condition, by contrast, is more subtle as it is not so much a change in the visual perception, but in a change of the social context of the situation. Humans might thus use fundamentally different cognitive mechanisms to switch between the individual and robot condition in particular.

In this section, we investigate how the trained network model differentiates the three conditions, looking specifically at change in activations of neurons in the recurrent layer while replicating data of the three conditions. Notably, differences between the experimental conditions are coded in our model only in terms of the behavior (i.e. the reproduced lengths). Differences in the way of presentation that were present in the human experiment (e.g. whether points appear on a screen or a robot touches the screen) were not explicitly modeled in the network. Thus, if we find that the network codes differences between the conditions differently in the three conditions, this indicates that these information must have been coded in the behavioral data of the human experiment, and the network automatically extracted them in order to solve the learning task.

Unlike experiment 1, this analysis does not require us to modify any hyperparameter. Instead, we directly observe how the network self-organizes its structure to accommodate the dynamics caused by the three different experimental conditions. Since these are all trained within the same network, we can directly compare their corresponding network dynamics. The core question is thus whether the network dynamics reflect the differences between the individual and the robot conditions and between the two robot conditions, respectively, in different ways.

We therefore investigate how different conditions are represented in the internal activations of the neurons of the neural networks over the course of the trajectory (i.e. from time step 0 to time step 21). The activations at one point in time are a 25-dimensional vector containing the activation values of all the neurons in the recurrent network layer of the internal model. These vectors were generated for each time step, and for all human experimental data, using the corresponding initial state of the participant and the condition in which the behavior was presented.

### 5.1 Results

An illustration of the network activations of time step 0 and time step 21 can be found in Fig 7. The activations are shown in the two-dimensional space generated via principal component analysis (PCA) from the original 25-dimensional vectors. In the left plot, the activations at time step 0 are shown, which correspond to the 75 initial states. colored symbols label different experimental conditions, the black symbols and ellipses show the mean and the covariance of the three conditions. The right plot shows the activations at time step 21. Note that more

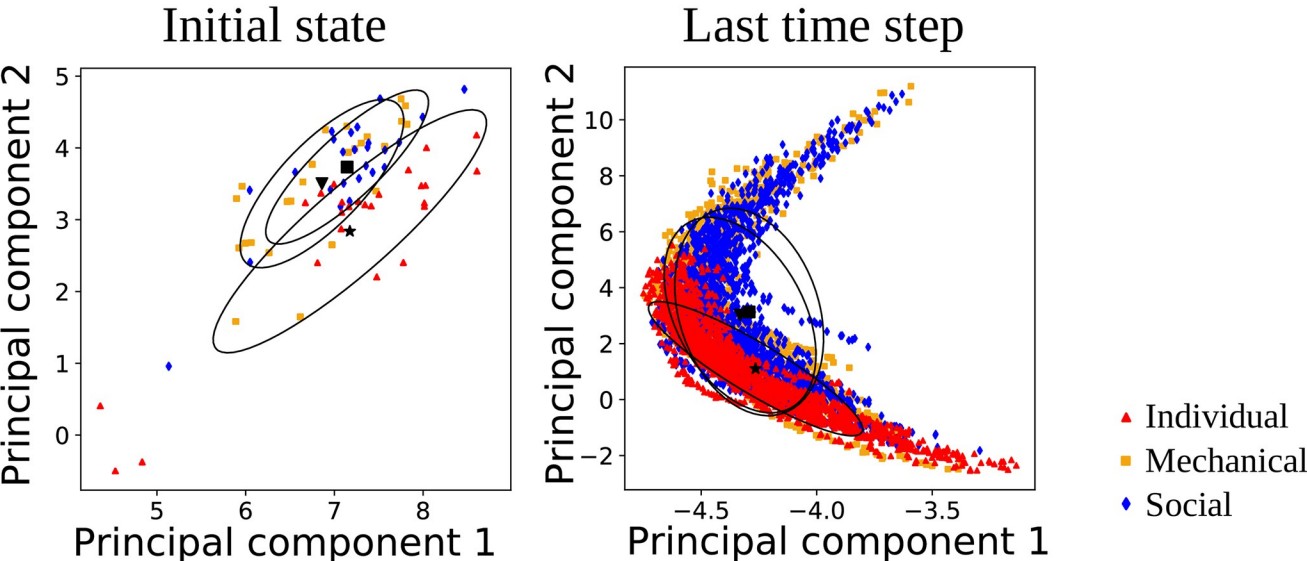

**Fig 7. The first two principal components of the network activation traces of one example network (capturing 83% of the variance), at the first time step (left) and at the last time step (right).** The black symbols show the mean, ellipses the covariances of the points of the corresponding experimental conditions.

points are visible in the right plot compared to the left plot because at $t = 0$ the trajectories still cannot be differentiated depending on their length whereas this differentiation is reflected in the network activations at $t = 21$.

Qualitatively, it can be observed that the mean and the covariances are similar for the mechanical and social robot conditions, in the first as well as in the last time step. This result is to be expected because the behavior in these two conditions was more similar to each other. However, a difference between the first and the last time step can be observed in the covariances: in the first time step, the covariance is larger in the individual condition than in the robot conditions, whereas in the last time step the covariance appears relatively smaller in the individual conditions.

This covariance indicates how variable the internal activations are in each of the three experimental conditions. A higher variability at time step $t$ indicates that the differences that arise between participants in this condition are coded more strongly in the network dynamics at this point in time.

Fig 7 shows only the results of a single trained network. To investigate whether there is a systematic change of variability over the course of time, we quantitatively measured the variability in the network activations of the three experimental conditions for all the ten networks across time.

To compute the variability between the activations of different participants in the network, we calculated the distances of the networks' activations within the three conditions as visualized in the scheme in Fig 8. In essence, activations were grouped into different categories depending on the length of the stimuli (eleven length categories were selected by identifying the most common presented lengths in the human data, namely, lengths which were presented more than 100 times during the experiment) and distances are computed only within the length categories. The reason for this procedure is that we want to measure the differences in how different participants are represented in the network, but *not* differences in the

## Example data (including 3 reproductions with different lengths for 3 participants):

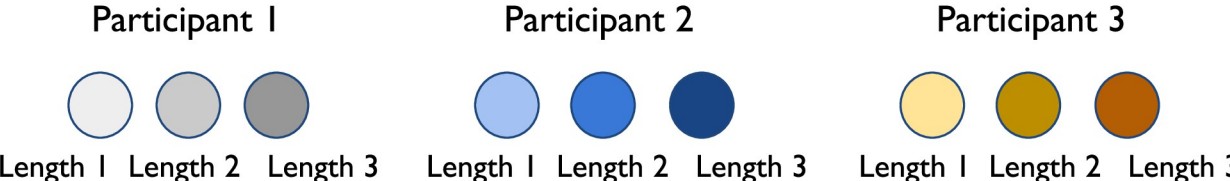

Participant 1 Participant 2 Participant 3

Length 1 Length 2 Length 3 Length 1 Length 2 Length 3 Length 1 Length 2 Length 3

## Evaluation of distances within lengths, across participants:

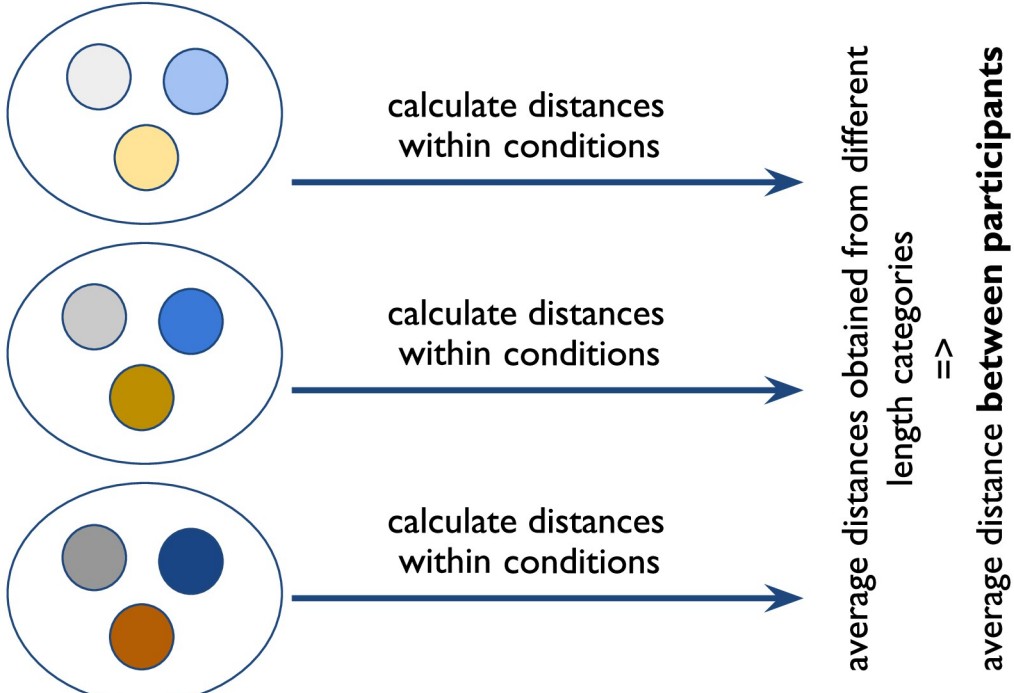

**Fig 8. Explanation of how the pairwise distances across participants were computed from the neural activation traces.** Each circle represents one trajectory of $25 \times 22$ where 25 is the number of neurons and 22 is the number of time steps. Data is split into 11 length categories and the pairwise distances within conditions are computed for each length category individually and later averaged, such that differences between lengths do not affect the final measure. The final measure, thus, shows for each time step the average distance between participants (cf. Fig 9).

reproduced lengths that also affect the network activations. Thus, the distances between all two activation vectors $\mathbf{x}$ and $\mathbf{y}$ of the same length category and experimental condition are computed as $1/N \cdot \sum_i (\sqrt{(x_i - y_i)^2})$. The results are shown in Fig 9. This plot shows the mean and standard error across the ten networks of the variability between activations of the same experimental condition. In line with the qualitative results in Fig 7, it can be observed that the individual condition has the highest variability in the beginning and the lowest variability in the end of trajectory generation.

The differences between the variability of the individual condition and the social condition are statistically significant in time step $t = 0$ (p < .05, $R^{m2}$ = .16) as well as in time step $t = 21$

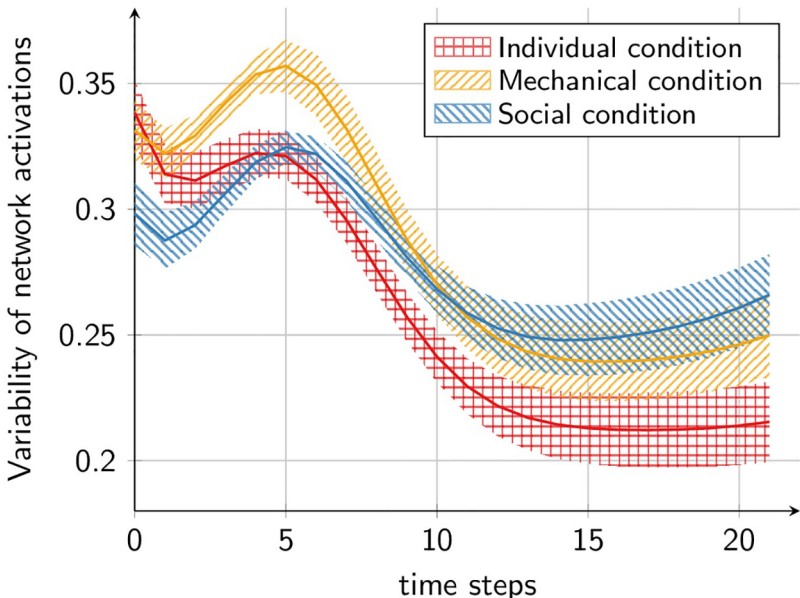

**Fig 9. Mean and standard error across networks of the average pairwise distances between the neural activation traces of the three different conditions (cf. Fig 8).** Activations were normalized to [0, 1] independently for each network beforehand.

($p < .05$, $R^2 = .16$) when modeling the reproduced distance with linear mixed effect models and condition as fixed effect and network ID as random effect.

## 5.2 Discussion

Results from this experiment provide access to the differences that exist among the individual, the mechanical and the social conditions in the generation of the trajectories. In particular, we suggest that the variability of network activations for the three conditions throughout the 22-time steps allows for a deeper understanding of how different the encoding of the network is across the three conditions for the entire generation process of the trajectories. Specifically, if the variability is high at time step 0, this indicates that the network mainly used differences in the encoding of the initial state of the network for differentiating the conditions when starting with the trajectory generation. On the contrary, if the variability is high at the last time step of the trajectory generation, this suggests that the differences between the conditions were mainly affected by the differences in the input data.

We observe that at the beginning, the individual condition shows higher variability than the social condition. In contrast, at the end, the variability is higher for the social as compared to the tablet condition. Therefore, for replicating the individual condition, the network mainly relies on information about the initial state, i.e., the network's prior information. In contrast, the social condition is affected more strongly by the input data that is presented during trajectory generation. This finding suggests that the neural network used different mechanisms to differentiate between the participants, depending on the condition.

Whereas experiment 1 demonstrated that the differences between the conditions could be explainable via a single unified mechanism, experiment 2 hints at that the network might have used two fundamentally different strategies to encode the individual vs. the social condition, indicating that also multiple distinct mechanisms could be at play. Firstly, the network relied

on the differences in the initial states. In the experiment, this strategy could correspond to using context information about the perceptual task (top-down strategy). The second mechanism that the network used was the input signal (bottom-up strategy). In the perceptual experiment, the sensory information was richer in the case of the the social situation compared to the individual condition (the robot's finger movements vs. a dot on the screen). It is, therefore, plausible that the participant relied more strongly on this richer information in the case of the social interaction. Interestingly, the network shows a similar trend using more information about the input signal for generating the trajectories in the social condition compared to the other conditions. This finding makes it plausible that such an interplay of two mechanisms could explain the behavioral differences. However, note that the input signal provided to the network input solely included the behavioral output, i.e., there was no difference in the richness of the signal in the computational study depending on the conditions. This limitation, however, is at the same time a strength of the computational model: the results hint at differences in the behavioral trajectories of the different although no confounding factors were present in the input signal. Still, it would be important in the future to verify this finding, extending the experiment to explicitly include factors such as the perceptual richness of the signal.

## 6 Conclusion

The aim of this study was to investigate how behavioral differences caused by differences in the social context can be replicated in a neural system, in order to generate hypotheses about the underlying cognitive mechanisms. For this purpose, we trained a neural network with human behavioral data of an experiment studying visual perception of space where three different conditions were tested ranging from an individual to a social task setting.

First, we demonstrated that the hyperparameters of the computational model that control the precision of the sensory and prior signal, respectively, can account for the differences among the experimental conditions (experiment 1). Specifically, we found that altering the precision of the prior as well as the precision of sensory input can replicate the behavioral differences between the three conditions: a stronger reliance on the prior, as well as a weaker reliance on sensory input, equally shifted the behavioral output of the network from the human behavior in the social condition towards the behavior in the individual condition, in line with the finding of [15] that participants tended more towards the mean in the individual condition. This finding makes it plausible that the same cognitive mechanism could be underlying the perceptual differences between the three conditions. Alternatively, different mechanisms could be intervening jointly in the same inferential process of perception.

The advantage of the network modeling study is that we can analyze the network's internal representation in order to understand how it performs the task at the level of neuron activities. Therefore, in a second experiment we analyzed how the differences between the conditions were coded in the neural dynamics of the network, independently of the current context. Therefore, in this second experiment we did not artificially modify the network's mechanics (as in experiment 1), but directly explored how the network internally represented and differentiated the three experimental conditions. The findings support the hypothesis of a plurality of phenomena affecting visual perception of space. We found that the variations between the three conditions emerged at different moments in time, suggesting that different mechanisms are at play. At the beginning of trajectory reproduction, more information about the non-social conditions affected the network representation. At the end of the reproduction, the representation is strongly driven by the differences in the social conditions, potentially due to the richer visual input that was present in this task.

The findings of this second experiment indicate that the balance between sensory and prior information which we demonstrated in experiment 1 only tells a part of the story. All three experimental conditions were differentiated in the neural encoding in ways that are intuitively explainable by the design of the human behavioral experiment (i.e. the richer sensory information provided in the robot conditions, see Section 5.2). The network solves the different conditions in different ways although it did not know what differentiated the individual and the robot conditions in the first place. This finding is interesting because it indicates that the human behavior alone was sufficient to let network dynamics emerge differently between the conditions, although the task design was exactly the same in all three conditions in the computational study.

As already emphasized in [15], "shared perception" indeed seems to be an important aspect affecting our perception of the world. In this context, the word "shared" might mean both "communicated or disclosed to others" or "held and experienced in common" (see [26]) and refers to the stimulus that was shared between the social robot and the study participant. Relying more on the shared sensory information instead of individual experiences in shared perception is an important prerequisite for solving a task in cooperation.

In this study, we could add to the findings of [15] thanks to the use of a computational model of which we cannot only observe the behavior, but also the internal dynamics of the network, that is, how the network came to the decisions it made. Specifically, the neural representation of the stimuli in the network allowed us to look into the time dynamics during the replication of the stimuli—something that remained hidden in the human behavioral experiment in [15]. The proposed model simplifies cognition significantly, but still might capture something important about shared perceptions, that is, how humans perceive their environment in a social context. The development of computational models for testing potential underlying mechanisms of specific behaviors found in human experiments, thus, may be an important means to form new hypotheses that may be tested in future experiments.

A further potential step for this research is to provide cognitive robotics with a computational model of shared perception. This can be developed on a robotic platform in order to endow it with human cognitive mechanisms of perception that can take into account three different parameters: the sensory information, the prior, and the sociality of the context which impacts on the balance between the other two parameters. Such socially perceiving robots might indeed be used for further experiments in human-robot interaction to understand which social mechanisms would strengthen or reduce the phenomenon of shared perception.

Also, it could be interesting to repeat the experiment of [15] while looking at the dynamic changes of human behavior, by either changing the task design to a dynamic task, or by tracking human behavior over a longer time window.

Another important direction of future research is to strengthen the connection of the computational study to the field of neuroscience. Such a stronger focus on computational studies for investigating neural phenomena is advancing significantly in recent decades, and a substantial amount of this work has focused on a topic that is also relevant for this study, namely, the relevance of top-down and bottom-up processing on human cognition [4, 5, 39]. Our analyses showed that human behavior in social context might be affected by the precision of sensory and prior information, and that two temporally separated mechanisms might be involved. Neurobiological studies are required to understand which precise neural mechanism are underlying such differences. There is in fact evidence of neurobiological differences that can be measured in the human brain in social context. The most prominent finding is that social context affects the concentration of neuromodulators in the human brain [36, 46]. Interestingly, neuromodulators also have been connected to the Bayesian framework. Specifically, studies suggest that neuromodulators might affect perception by changing the reliance on

prior and sensory information [5, 37]. Nevertheless, our study can only provide a potential explanation but not verify the neurophysiological plausibility at this point. Further investigations are required to better understand the neurobiology underlying social behavior in the context of a task like the one we investigate for gaining deeper insights into cognitive mechanisms of shared perception.

## Supporting information

**S1 Appendix. The effect of Bayesian inference on the integration of sensory and prior information.** See Appendix for detailed description of the effect of Bayesian inference on the integration of sensory and prior information.
(PDF)

## Author Contributions

**Conceptualization:** Maria Tsfasman, Anja Philippsen, Yukie Nagai.

**Data curation:** Maria Tsfasman, Carlo Mazzola, Alessandra Sciutti.

**Formal analysis:** Maria Tsfasman.

**Funding acquisition:** Yukie Nagai.

**Investigation:** Maria Tsfasman, Anja Philippsen.

**Methodology:** Maria Tsfasman, Anja Philippsen.

**Project administration:** Anja Philippsen, Serge Thill, Yukie Nagai.

**Resources:** Carlo Mazzola.

**Software:** Maria Tsfasman.

**Supervision:** Anja Philippsen, Serge Thill, Yukie Nagai.

**Validation:** Anja Philippsen.

**Visualization:** Maria Tsfasman, Anja Philippsen.

**Writing – original draft:** Maria Tsfasman, Carlo Mazzola.

**Writing – review & editing:** Maria Tsfasman, Anja Philippsen, Carlo Mazzola, Serge Thill, Alessandra Sciutti, Yukie Nagai.

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
