## [Decision Letter · Decision Letter 0]

2 May 2022

PONE-D-22-06499The world seems different in a social context: a neural network analysis of human experimental dataPLOS ONE

Dear Dr. Tsfasman,

Thank you for submitting your manuscript to PLOS ONE. After careful consideration, we feel that it has merit but does not fully meet PLOS ONE’s publication criteria as it currently stands. Therefore, we invite you to submit a revised version of the manuscript that addresses the points raised during the review process.

We look forward to receiving your revised manuscript.

Kind regards,

Kiyoshi Nakahara, PhD

Academic Editor

PLOS ONE

Journal Requirements:

“This research was supported by JST CREST ‘Cognitive Mirroring’ (Grant Number: JPMJCR16E2), Institute for AI and Beyond at the University of Tokyo, and World Premier International Research Center Initiative (WPI), MEXT, Japan and by the Starting Grant wHiSPER (G.A. No 804388) from the European Research Council (ERC) under the European Union’s Horizon 2020 research and innovation programme.”

“This research was supported by JST CREST ‘Cognitive Mirroring’ (Grant Number: JPMJCR16E2), Institute for AI and Beyond at the University of Tokyo, and World Premier International Research Center Initiative (WPI), MEXT, Japan and by the Starting Grant wHiSPER (G.A. No 804388) from the European Research Council (ERC) under the European Union’s Horizon 2020 research and innovation programme.”

“This research was supported by JST CREST ‘Cognitive Mirroring’ (Grant Number: JPMJCR16E2), Institute for AI and Beyond at the University of Tokyo, and World Premier International Research Center Initiative (WPI), MEXT, Japan and by the Starting Grant wHiSPER (G.A. No 804388) from the European Research Council (ERC) under the European Union’s Horizon 2020 research and innovation programme.”

Reviewers' comments:

Reviewer's Responses to Questions

**Comments to the Author**

1. Is the manuscript technically sound, and do the data support the conclusions?

Reviewer #1: Partly

2. Has the statistical analysis been performed appropriately and rigorously? 

Reviewer #1: No

3. Have the authors made all data underlying the findings in their manuscript fully available?

Reviewer #1: Yes

4. Is the manuscript presented in an intelligible fashion and written in standard English?

Reviewer #1: Yes

5. Review Comments to the Author

Reviewer #1: This paper investigates the mechanism behind the social context-dependence of the central tendency effect by training a model that combines Bayesian inference and processing in neural networks using data from the previous study in the field of human-agent interaction. The conclusion that the degree of reliance on prior distributions explains the results observed in the previous study is particularly interesting. However, I consider that the validity of the conclusions of the present study is limited for the following reasons:

1. The original experiment was poorly controlled for confounding compared to psychological standards, and it is questionable that it can be considered an experiment showing a relationship between the central tendency effect and social natures.

2. The cognitive neuroscience and neurophysiological relevance of the proposed model is unclear.

3. Insufficient explanation of the architecture and mathematical mechanism of the proposed model makes it difficult to understand the model validity.

4. It is unclear what aspects of the experiment are reflected by the parameters manipulated in the analysis.

5. The discussion connecting the results and conclusions is too speculative.

Comments:

Overall

The experiment in the ref. 15 that provides data used in the present study was not well controlled for confounding compared to psychological standards. This means that it is difficult to consider that the study in the ref. 15 investigated the influence of the social natures of the contexts on the central tendency effect.

It is unclear that the logic linking the manipulation of the precision of the sensory and the prior signals considered in this study with the manipulation of social contexts addressed in ref. 14. The authors showed that changes in the values of the reliance parameters H_prior and H_sensor modulated the central tendency effects. But how does it conceptually relate to the experimental conditions in the ref. 15? What aspects of the experimental conditions are reflected in the small or large values of the H_prior and H_sensor?

In experiment 1, the authors have succeeded in reproducing this difference in the three experimental conditions by varying the value of the parameter Hs. However, this is only a sufficient condition, and this result does not indicate that this is the physiological mechanism working in the brain that produced the results of this experiment. To demonstrate the validity of the mechanism proposed by the authors, it is first necessary to show that the proposed model is physiologically valid, citing the results of neurophysiological studies. However, this has not been done sufficiently.

Section 1

In line 17, it is unclear what the authors express by the word ‘prior’. Since the precision of the signals was learned from the experience, it should be represented as some prior distributions.

In line 37 (also in lines 82 and 88), the authors wrote as “these findings could be explained by differences in their reliance on priors: the closer subjects tended to the mean, the more they relied on prior experience (high central tendency); the closer they stick to the specific sensory input, the lower was their prior reliance (low central tendency).” However, this may be interpreted as that the subjects showing higher central tendency just have a broader prior distribution.

In line 45, although the findings in the present and the ref. 15 studies might be explainable based on some social factors, they also result from other factors such as intensities of the attention, cognitive loads, distances between the subject’s eyes and the screen, gaze directions, task complexity, and so on.

Section 2.2

Please show more details of the architecture of your S-CTRNN model such as the number of units in each layer, activations functions, connections, etc. Also, the authors should describe the mathematical formulae of the L_init and the Bayesian inference module more rigorously. In addition, the variables and parameters should be described more precisely. For example, are they scalars, vectors, or matrices? What are their domains of definition?

In line 202, I do not know whether the way adopted by the authors in which different participants are expressed as the different values of the initial states of a network is reasonable. In the fifth section, the authors tried to extract some insights into the neural processing underlying the conditional dependence of the central tendency effects by analyzing the network as a model of the biological neural circuits. However, an actual biological brain is not shared among the subjects.

Figure 1 and Section 3.1.1

It is difficult to know how the data explained here were actually transformed into the inputs described in Fig. 1. I guess that the variables shown as x_t and sigma^2_sensor in the figure were generated from the data. However, I cannot understand how you did it. Please explain the way by using the symbols that appeared in Fig. 1 (e.g. x_t and sigma^2_sensor). Additionally, please make the caption of the figure more informative.

Section 3.2

In line 330, please show not only p-values but also test statistics, sample sizes, and confidence intervals.

Sections 4.1 and 4.2 and Figs. 5 and 6

In these sections and the figures, the results for different values of the parameter H_prior or H_sensor in a trained network under social conditions were shown. I think that the results of varying the parameter Hs in networks trained under the other conditions should also be shown. It would be helpful to the reader if it were included somewhere.

Section 5

In line 525, does this mean that the networks directly trained for each condition are the ones analyzed here? The most important concern in this paper is the dependence of the central tendency effects on parameter Hs. Therefore, the same analysis should be performed for the case where the behavior observed in the human experiment is reproduced by adjusting the parameter Hs (i.e. in the case of experiment 1).

Sections 5.2 and 6

In the second paragraph and the following in section 5.2 and the part of section 6 discussing the results of experiment 2, the argument sounds very speculative. First, the model does not incorporate enough factors that could affect the experimental results to make it convincing that the relationship between the steps and the variabilities in each condition observed in the model also occurs in the actual participant's brain. While the large variability in the first step may well be dependent on the initial condition, in a real experiment the social condition is likely to be confounded by factors of perception, attention, and cognitive load. In the present model, at least, this confounding is not reflected in the inputs or initial conditions. But for this argument to be convincing, it is necessary to test whether or not this is reflected in the inputs or initial conditions do not affect the results. Second, the argument that the variability in the last step reflects differences in sensory signals due to experimental conditions is difficult to understand. Given that the states at the last steps are directly related to the final outputs, it is natural to consider that the variability at the last step reflects the variability of the teacher values applied to the outputs.

6. PLOS authors have the option to publish the peer review history of their article (what does this mean?). If published, this will include your full peer review and any attached files.

Reviewer #1: No

---

## [Author Response · Author response to Decision Letter 0]

28 Jul 2022

Dear Reviewer,

Thank you for taking the time to review our manuscript and for your helpful suggestions.

We addressed your comments carefully in the revised manuscript as detailed below.

In the following, we list the reviewer’s comments in gray, and our reply in blue color (better viewed in the pdf version of the response):

Reviewer #1: This paper investigates the mechanism behind the social context-dependence of the central tendency effect by training a model that combines Bayesian inference and processing in neural networks using data from the previous study in the field of human-agent interaction. The conclusion that the degree of reliance on prior distributions explains the results observed in the previous study is particularly interesting. However, I consider that the validity of the conclusions of the present study is limited for the following reasons:

1. The original experiment was poorly controlled for confounding compared to psychological standards, and it is questionable that it can be considered an experiment showing a relationship between the central tendency effect and social natures.

We adjusted the description of the original experiment to indicate that while the comparison between the individual vs. robot condition did not allow for controlling all confounding factors, the manipulation that was designed for the comparison between the mechanical and social robot conditions, was carefully controlled, to guarantee the presentation of the same distance stimuli, while maintaining a realistic interaction setting. The results of such a comparison can then provide sufficient evidence for the phenomenon investigated in this study. See also reply below: [Reply1] 

2. The cognitive neuroscience and neurophysiological relevance of the proposed model is unclear.

We addressed this comment by strengthening the connection between neuroscientific evidence on how perception changes in social situations and the predictive coding view on the central tendency effects. See replies below [Reply2, Reply3].

3. Insufficient explanation of the architecture and mathematical mechanism of the proposed model makes it difficult to understand the model validity.

We extended the explanation in the manuscript and also added the mathematical formulas as suggested. See reply below [Reply7].

4. It is unclear what aspects of the experiment are reflected by the parameters manipulated in the analysis.

By more clearly explaining how social factors are hypothesized to affect perceptual phenomenons, we hope that these connection becomes clearer, see reply below: [Reply2]

5. The discussion connecting the results and conclusions is too speculative.

See reply below [Reply13].

Comments:

Overall

“The experiment in the ref. 15 that provides data used in the present study was not well controlled for confounding compared to psychological standards. This means that it is difficult to consider that the study in the ref. 15 investigated the influence of the social natures of the contexts on the central tendency effect.”

[Reply1] Regarding the differences between the “individual” vs. “with-robot” conditions, the reviewer is right that a direct comparison needs to be performed with care. Indeed, the individual condition was designed mainly as a baseline to assess the basic degree of central tendency of each participant with a paradigm traditionally adopted in visual perception studies.This is why the main value of that paper is the comparison between the two robot-based conditions. These conditions were carefully controlled: they are identical in terms of spatial configuration of the participant and the robot, their relative position with respect to the screen and in the properties of the arm motion showing the distances to be reproduced. The only difference between these conditions was the experimental manipulation (e.g., the gaze behavior and the speech of the robot) that was expected to modify the context of the situation from “with a machine” to “with another agent”, i.e. social. 

Of course it may be discussed which aspect of the “social” context actually induced the effect in the participants (i.e., the behavior of the robot, or the knowledge of being involved in an interaction). However, the significant correlation observed between the perceived anthropomorphism in the social condition and the amount of change in context dependency excludes the possibility that the social condition was simply in general richer or more informative. Rather, it points to the effectiveness of the manipulation as a determinant factor in the modification of the regression to the mean phenomenon.

Overall, we believe that the differences between the mechanical and the social robot condition can be taken as evidence that there is an effect of the social context on the participant’s perception as measured by the central tendency task.

We added this explanation to Section 2.1. (see lines 147-152, 179-183, 184-186 of the manuscript)

“It is unclear that the logic linking the manipulation of the precision of the sensory and the prior signals considered in this study with the manipulation of social contexts addressed in ref. 14. The authors showed that changes in the values of the reliance parameters H_prior and H_sensor modulated the central tendency effects. But how does it conceptually relate to the experimental conditions in the ref. 15? What aspects of the experimental conditions are reflected in the small or large values of the H_prior and H_sensor?”

[Reply2] The suggestion that Bayesian mechanisms, specifically predictive coding, might explain the central tendency effect was not originally proposed by us but has been frequently made in behavioral and neurophysiological literature [e.g., "Predictive coding of multisensory timing" Shi et al., Current Opinion in Behavioral Sciences, 2016 or "When the world becomes too real: a Bayesian explanation of autistic perception", Pellicano & Burr].

We demonstrate that it is in fact possible to replicate the central tendency effect by changing the relative precision of sensory and prior information based on the data collected in the experiment of reference 15 in an experiment manipulating the social context.

As the social context was shown to affect the central tendency effect, changes on the reliance on prior and sensory information can be considered as a potential neurological mechanisms mediating the change in perception. While this remains of course a theory that can not be confirmed by this computational study alone, we demonstrate that such a mechanism offers a potential explanation.

Literature shows that this mechanism also appears neurophysiologically plausible. Specifically, there have been studies showing that neuromodulators, which play an important role in social behavior, have an effect on human perception, specifically, on changing the sensory precision [e.g., "From drugs to deprivation: a Bayesian framework for understanding models of psychosis", Corlett et al. Psychopharmacology volume 206, 2009].

Also alternative mechanisms could be imagined, for instance, attention. Social interaction could highlight signals from the external world and, thus, cause participants to attend more strongly to sensory information which are more salient (i.e., more “precise”).

In fact, there is only limited evidence for now to support the above mentioned mechanisms. However, our intention is to propose new hypotheses to be tested in neuroscience and/or psychological experiments in the future. This paper is, thus, valuable in encouraging new experimental designs.

We revised the introduction to make this connection between neurobiological evidence and our experimental design more explicit. (see pages 2 and 3 in the manuscript)

“In experiment 1, the authors have succeeded in reproducing this difference in the three experimental conditions by varying the value of the parameter Hs. However, this is only a sufficient condition, and this result does not indicate that this is the physiological mechanism working in the brain that produced the results of this experiment. To demonstrate the validity of the mechanism proposed by the authors, it is first necessary to show that the proposed model is physiologically valid, citing the results of neurophysiological studies. However, this has not been done sufficiently.”

[Reply3]

As mentioned above (see [Reply2]), our main intention in this study is not to verify existing neurophysiological evidences with computational approaches but to use a computational model to propose new computational hypotheses to be tested in neural psychophysical experiments in the future.

Although there exists only limited neurophysiological evidence in the literature to support the mechanisms that we are highlighting in this study, we believe that demonstrating the feasibility is still important because it might motivate further neurophysiological studies into this direction.

At the same time, the mechanism is based on a well accepted theory of how perception works in the framework of predictive coding. Also, there exists at least initial evidence indicating that neuromodulators are a potential neurophysiological mechanism that may cause changes in the perception similar to the ones that we simulate via the H parameter. We discuss this now in more detail in the introduction.

Furthermore, we added a disclaimer in the conclusion section stating that our study can only provide a potential explanation but not verify the neurophysiological plausibility. (see lines 763-767 in the manuscript)

Section 1

“In line 17, it is unclear what the authors express by the word ‘prior’. Since the precision of the signals was learned from the experience, it should be represented as some prior distributions.”

[Reply4] We replaced “prior” with “prior distribution in the referenced sentence to indicate that it is a distribution. (see line 14 of the manuscript)

“In line 37 (also in lines 82 and 88), the authors wrote as “these findings could be explained by differences in their reliance on priors: the closer subjects tended to the mean, the more they relied on prior experience (high central tendency); the closer they stick to the specific sensory input, the lower was their prior reliance (low central tendency).” However, this may be interpreted as that the subjects showing higher central tendency just have a broader prior distribution.”

[Reply5] It is true that the same result could be explained by two different parameter changes: a broader prior distribution or a more precise sensory distribution both would reduce the reliance on the prior and, thus, reduce the central tendency effect.

We changed the description to indicate that both hypotheses are valid as it has been also indicated in the literature [e.g., "Alternative Bayesian accounts of autistic perception: comment on Pellicano and Burr.", Brock, Jon.Trends in cognitive sciences, 2012].

In this study, we are interested in understanding the relative difference between the two precision terms. Determining which mechanism would be neurophysiologically more realistic is out of scope of the current study. (see lines 38-51 of the manuscript)

“In line 45, although the findings in the present and the ref. 15 studies might be explainable based on some social factors, they also result from other factors such as intensities of the attention, cognitive loads, distances between the subject’s eyes and the screen, gaze directions, task complexity, and so on.”

[Reply6]

The reviewer is right that differences in the behavior based on a social situation is not directly caused by the situation "being social", but mediated by neurobiological and cognitive mechanisms such as attention. We now clarify the connection to attention in the introduction (see also [Reply2]).

Regarding other differences in the task, the original experiment was carefully controlled to exclude factors such as task complexity and cognitive load between the mechanical and the social condition. But the individual condition could indeed differ from the other two conditions with respect to such factors (see also [Reply1]). We adjusted the sentence in the mentioned line slightly to take this into account.

Section 2.2

“Please show more details of the architecture of your S-CTRNN model such as the number of units in each layer, activations functions, connections, etc. Also, the authors should describe the mathematical formulae of the L_init and the Bayesian inference module more rigorously. In addition, the variables and parameters should be described more precisely. For example, are they scalars, vectors, or matrices? What are their domains of definition?”

[Reply7] We added the formula for L_init and information on the number of context layers and connections between the layers. We also explicitly explain now the used notation and the domains of definition. (See the description of the model in Section 2.2., lines 207-211, 222-228 of the manuscript)

The formulas for the Bayesian inference module are included as Eq. (3) and (4) in the new version of the document. 

“In line 202, I do not know whether the way adopted by the authors in which different participants are expressed as the different values of the initial states of a network is reasonable. In the fifth section, the authors tried to extract some insights into the neural processing underlying the conditional dependence of the central tendency effects by analyzing the network as a model of the biological neural circuits. However, an actual biological brain is not shared among the subjects.”

[Reply8]

Using different initial states in a recurrent neural network separates the network into different subnetworks which we use to split between conditions and subjects. Importantly, the neural network here is not intended to be biologically plausible in the sense that it simulates the mechanisms happening in the brain of the participant. Instead, we use it as a computational tool. Here, summarizing all participants and conditions in one network is convenient as it allows us to analyze the differences between all conditions and participants under fair conditions (i.e., independently of divergence between networks that may result from training). Furthermore, this design allows us to demonstrate that the H parameter alone is sufficient to switch between different conditions in Section 4.

Also, we believe that a significant overlap between the used neuron connection is unlikely, given the relatively high number of neurons (N=25) which are all interconnected (25*25 = 225 connections), and the low complexity of the 1-dimensional task.

We clarified this now in the text.

Figure 1 and Section 3.1.1

“It is difficult to know how the data explained here were actually transformed into the inputs described in Fig. 1. I guess that the variables shown as x_t and sigma^2_sensor in the figure were generated from the data. However, I cannot understand how you did it. Please explain the way by using the symbols that appeared in Fig. 1 (e.g. x_t and sigma^2_sensor). Additionally, please make the caption of the figure more informative.”

[Reply9]

Each point in the input signal (blue points on the left-hand side of the figure) is x_t, sigma^2_sensor is the variance that is associated with the input signal. 

Section 3.2

“In line 330, please show not only p-values but also test statistics, sample sizes, and confidence intervals.”

[Reply10]

We updated Figure 4 to display not only p-values but also chi-square statistics.

The caption was updated to indicate the confidence intervals and the sample size.

Sections 4.1 and 4.2 and Figs. 5 and 6

“In these sections and the figures, the results for different values of the parameter H_prior or H_sensor in a trained network under social conditions were shown. I think that the results of varying the parameter Hs in networks trained under the other conditions should also be shown. It would be helpful to the reader if it were included somewhere.”

[Reply11]

The networks were trained not only with the social condition but including the participant data from all conditions with different initial states reflecting different participants and different conditions.

Probably the reviewer refers to the fact that we only show the results for replicating the mechanical and the individual condition, starting from the initial states of the social condition, and do not test other combinations.

As we argued in Section 4.1, we chose to test the shift from a weak to a strong prior because the networks were trained to replicate the human data but do not have access to the actual presented data. Therefore, the model cannot replicate data more precisely even if the precision of sensory data as opposed to prior information is reduced. As a result, the initial states of the tablet condition replicate the human data equally for H=1 as for H close to 0 (using only minimal prior information), as shown in this figure:

In contrast, moving from a weaker to a stronger prior is possible because the mean of the data is implicitly known to the network.

To make this result more explicit, we now report the experimental results also for shifting from a stronger to a weaker prior, to illustrate that this shift cannot be demonstrated using the current experimental design. The figure will be made available at publication time in the corresponding github repository.

Section 5

“In line 525, does this mean that the networks directly trained for each condition are the ones analyzed here? The most important concern in this paper is the dependence of the central tendency effects on parameter Hs. Therefore, the same analysis should be performed for the case where the behavior observed in the human experiment is reproduced by adjusting the parameter Hs (i.e. in the case of experiment 1).”

[Reply12]

There might have been a misconception about the aim of this second experiment which was to observe the self-organization of the neural network during training to incorporate the differences between the conditions and the different participants as obtained during training.

Hereby, we are interested in investigating the mechanisms that the neural network uses internally to differentiate the conditions, that means, independently of the parameter H.

If we would change the parameter H during this analysis we would be able to measure differences between the conditions, but those would be differences induced by our manipulations in the first place, not by the original network training, leading to a circular conclusion.

Therefore, we believe that the analysis is more interesting in its current forms as it provides insights into how the neural network self-organizes the neural activations to accommodate the different conditions. This comparative analysis is possible as we trained all subjects and conditions in a single network (see [Reply8]).

Furthermore, the initial states which formed already during training would not be affected by such a change in the H parameter at run time, leading to an unfair comparison between the first and the last timestep.

We adjusted the description of the experiments in the introduction and in the conclusion to make this difference between the experiments clearer.

Sections 5.2 and 6

“In the second paragraph and the following in section 5.2 and the part of section 6 discussing the results of experiment 2, the argument sounds very speculative. First, the model does not incorporate enough factors that could affect the experimental results to make it convincing that the relationship between the steps and the variabilities in each condition observed in the model also occurs in the actual participant's brain. While the large variability in the first step may well be dependent on the initial condition, in a real experiment the social condition is likely to be confounded by factors of perception, attention, and cognitive load. In the present model, at least, this confounding is not reflected in the inputs or initial conditions. But for this argument to be convincing, it is necessary to test whether or not this is reflected in the inputs or initial conditions do not affect the results. Second, the argument that the variability in the last step reflects differences in sensory signals due to experimental conditions is difficult to understand. Given that the states at the last steps are directly related to the final outputs, it is natural to consider that the variability at the last step reflects the variability of the teacher values applied to the outputs.”

[Reply13]

The fact that the "sensory input" provided to the model is different from the real sensory input is an important point that was not sufficiently emphasized in the previous version of the manuscript. We thank the reviewer for pointing this out.

Our aim in Experiment 2 is to observe the mechanisms that the computational model uses for differentiating the different conditions. We do not claim that the same mechanisms are at play in the participant's brain.

To avoid such a misunderstanding, we restructured the discussion to separate more carefully the description of the results at a technical level and the potential conclusions that we could draw from this finding.

Specifically, first, we focus on answering the question of what the results tell us about the computational mechanisms that the neural network model relies on for differentiating the conditions. In a second step, we discuss what these findings could implicate for the perceptual study, while mentioning the limitations of this comparison.

---

## [Decision Letter · Decision Letter 1]

15 Aug 2022

The world seems different in a social context: a neural network analysis of human experimental data

PONE-D-22-06499R1

Dear Dr. Tsfasman,

We’re pleased to inform you that your manuscript has been judged scientifically suitable for publication and will be formally accepted for publication once it meets all outstanding technical requirements.

Kind regards,

Kiyoshi Nakahara, PhD

Academic Editor

PLOS ONE

Additional Editor Comments (optional):

Reviewers' comments:

Reviewer's Responses to Questions

**Comments to the Author**

1. If the authors have adequately addressed your comments raised in a previous round of review and you feel that this manuscript is now acceptable for publication, you may indicate that here to bypass the “Comments to the Author” section, enter your conflict of interest statement in the “Confidential to Editor” section, and submit your "Accept" recommendation.

Reviewer #1: All comments have been addressed

2. Is the manuscript technically sound, and do the data support the conclusions?

Reviewer #1: Yes

3. Has the statistical analysis been performed appropriately and rigorously? 

Reviewer #1: Yes

4. Have the authors made all data underlying the findings in their manuscript fully available?

Reviewer #1: Yes

5. Is the manuscript presented in an intelligible fashion and written in standard English?

Reviewer #1: Yes

6. Review Comments to the Author

Reviewer #1: (No Response)

7. PLOS authors have the option to publish the peer review history of their article (what does this mean?). If published, this will include your full peer review and any attached files.

Reviewer #1: **Yes: **Hiroki Kurashige

---

## [Editor Report · Acceptance letter]

22 Aug 2022

PONE-D-22-06499R1 

The world seems different in a social context: a neural network analysis of human experimental data 

Dear Dr. Tsfasman:

I'm pleased to inform you that your manuscript has been deemed suitable for publication in PLOS ONE. Congratulations! Your manuscript is now with our production department. 

Kind regards, 

on behalf of

Dr. Kiyoshi Nakahara 

Academic Editor

PLOS ONE